# Adapting the Fitness Criteria for Non-Intensive Treatments in Older Patients with Acute Myeloid Leukemia to the Use of Venetoclax-Hypomethylating Agents Combination—Practical Considerations from the Real-Life Experience of the Hematologists of the Rete Ematologica Lombarda

**DOI:** 10.3390/cancers16020386

**Published:** 2024-01-16

**Authors:** Giuseppe Rossi, Erika Borlenghi, Patrizia Zappasodi, Federico Lussana, Massimo Bernardi, Claudia Basilico, Alfredo Molteni, Ivana Lotesoriere, Mauro Turrini, Marco Frigeni, Monica Fumagalli, Paola Cozzi, Federica Gigli, Chiara Cattaneo, Nicola Stefano Fracchiolla, Marta Riva, Gianluca Martini, Valentina Mancini, Roberto Cairoli, Elisabetta Todisco

**Affiliations:** 1Department of Hematology, ASST Spedali Civili of Brescia, 25100 Brescia, Italy; erika.borlenghi@gmail.com (E.B.); chiara.cattaneo@asst-spedalicivili.it (C.C.); 2Division of Hematology, Fondazione IRCCS Policlinico San Matteo, University of Pavia, 27100 Pavia, Italy; p.zappasodi@smatteo.pv.it (P.Z.); gianluca.martini01@universitadipavia.it (G.M.); 3Department of Oncology and Hematology, University of Milan and Azienda Socio Sanitaria Territoriale Papa Giovanni XXIII, 24100 Bergamo, Italy; flussana@asst-pg23.it (F.L.); mfrigeni@asst-pg23.it (M.F.); 4Haematology and Bone Marrow Transplantation Unit, San Raffaele Scientific Institute, 20132 Milano, Italy; bernardi.massimo@hrs.it; 5Division of Hematology, ASST Sette Laghi, Ospedale di Circolo e Fondazione Macchi, 21100 Varese, Italy; claudiamaria.basilico@asst-settelaghi.it; 6Department of Hematology, ASST di Cremona, 26100 Cremona, Italy; alfredo.molteni@asst-cremona.it; 7Department of Hematology, Ospedale Busto Arsizio, ASST Valle Olona, 21052 Busto Arsizio, Italy; ivana.lotesoriere@asst-valleolona.it; 8Division of Hematology, Ospedale Valduce, 22100 Como, Italy; mturrini@valduce.it (M.T.); elisabetta.todisco@asst-valleolona.it (E.T.); 9Department of Hematology, Ospedale “San Gerardo” ASST di Monza, 20900 Monza, Italy; m.fumagalli@asst-monza.it; 10Unità Complessa di Ematologia, ASST Ovest Milanese, Ospedale Civile, 20025 Legnano, Italy; paola.cozzi@asst-ovestmi.it; 11Divisione di Emato-Oncologia, European Institute of Oncology, 20122 Milano, Italy; federica.gigli@ieo.it; 12UOC Oncoematologia, Fondazione IRCCS Ca’ Granda Ospedale Maggiore Policlinico and University of Milan, 20122 Milano, Italy; ns.fracchiolla@gmail.com; 13Dipartimento di Ematologia ed Oncologia, Niguarda Cancer Center ASST Grande Ospedale Metropolitano, 20162 Milano, Italy; marta.riva@ospedaleniguarda.it (M.R.); valentina.mancini@ospedaleniguarda.it (V.M.); roberto.cairoli@ospedaleniguarda.it (R.C.)

**Keywords:** fitness, elderly, acute myeloid leukemia, treatment

## Abstract

**Simple Summary:**

In older AML patients, their clinical fitness is of utmost importance for choosing the most appropriate therapy. Therefore, treatment-specific fitness criteria were devised in 2013 by SIE/SIES/GITMO to select patients deemed unfit for intensive chemotherapy (ICT) or even hypomethylating agents (HMAs). Since then, the therapeutic armamentarium for patients unfit for ICT has been enriched. In the present analysis of over 500 patients treated in REL centres, venetoclax/HMAs emerged as the most frequently used treatment. Considering its unique toxicity profile, an update of treatment-specific fitness criteria for selecting candidates for venetoclax/HMAs would be desirable. REL hematologists, who have gained experience with the combination over the last years, were asked if they actually restrict SIE/SIES/GITMO fitness criteria for HMAs when candidating patients to venetoclax/HMAs. A broad consensus emerged on limiting its choice to patients younger than 80–85, with a cardiac EF > 40%, without significant pulmonary comorbidities, and with an adequate caregiver.

**Abstract:**

A retrospective survey was conducted in hematologic centres of the Rete Ematologica Lombarda (REL) on 529 older AML patients seen between 2020–2022. Compared to 2008–2016, the use of intensive chemotherapy (ICT) decreased from 40% to 18.1% and of hypomethylating agents (HMAs) from 19.5% to 13%, whereas the combination of Venetoclax/HMA, initially not available, increased from 0% to 36.7%. Objective treatment-specific fitness criteria proposed by SIE/SIES/GITMO in 2013 allow an appropriate choice between ICT and HMAs by balancing their efficacy and toxicity. Venetoclax/HMA, registered for patients unfit to ICT, has a unique toxicity profile because of prolonged granulocytopenia and increased infectious risk. Aiming at defining specific fitness criteria for the safe use of Venetoclax/HMA, a preliminary investigation was conducted among expert REL hematologists, asking for modifications of SIE/SIES/GITMO criteria they used to select candidates for Venetoclax/HMA. While opinions among experts varied, a general consensus emerged on restricting SIE/SIES/GITMO criteria for ICT-unfit patients to an age limit of 80–85, cardiac function > 40%, and absence of recurrent lung infections, bronchiectasis, or exacerbating COPD. Also, the presence of an adequate caregiver was considered mandatory. Such expert opinions may be clinically useful and may be considered when treatment-specific fitness criteria are updated to include Venetoclax/HMA.

## 1. Introduction

Acute myeloid leukemia (AML) is the most common acute leukemia subtype and has the poorest prognosis [1]. The prevalence of AML is highly dependent on age and increases substantially after 55 years of age [1,2]. Also, its prognosis worsens significantly with increasing age [2,3]. The 5-year overall survival (OS) of patients with AML is <25% and <10% in patients 60 to 65 and ≥70 years old, respectively, compared with 50% for those <50 years [1,4].

Traditional treatment strategies included chemotherapy, less-intensive therapies with hypomethylating agents (HMAs), or best supportive care (BSC) [5]. Apart from intensive chemotherapy, other treatment modalities have few probabilities of significantly affecting OS or even obtaining long-term disease-free survival in those patients who are not considered able to tolerate intensive chemotherapy. In recent years, a number of novel agents have been identified with biological targets and mechanisms of action different from cytostatic drugs, as well as with different and more manageable toxicity profiles, which have demonstrated significant activity, both alone and in combination, and have therefore completely changed the treatment paradigm in AML, particularly in patients unfit to intensive chemotherapy [6].

Age has historically been considered the main criterion to determine eligibility for intensive chemotherapy in patients with AML. Assessment of comorbidities and performance status and, more recently, comprehensive geriatric assessment has also been used to define the overall fitness of patients with the main purpose of identifying those not able to tolerate intensive chemotherapy. General prognostic scores, which combine clinical and hematological parameters with geriatric variables and comorbidity burden and efficiently identify subgroups of patients with markedly different prognoses, have been reported [7,8]. These scores were most frequently built on a series of patients treated with intensive chemotherapy, while few papers also dealt with patients receiving HMAs [9,10]. While these scores are important to help in the selection of the most effective treatment for the patients, they seldom identify specific parameters associated with the tolerance of patients to different therapeutic options, which could be useful to further tailor treatment to the clinical characteristics of the older patient. 

Among multiparameter tools, the Italian SIE/SIES/GITMO Consensus Criteria were the first to link fitness assessment to the specific therapeutic modalities available [11]. They defined criteria to be eligible for intensive chemotherapy, for non-intensive therapy, or just for best supportive care in older patients with AML. They were devised on the basis of the clinical experience of a panel of experts but were subsequently validated as a tool to predict the most appropriate treatment option regarding treatment-related toxicity and 100-d survival as well as overall long-term survival [12,13]. They represent a useful tool for hematologists in daily clinical practice and have also been used to objectively define unfitness to intensive chemotherapy in prospective clinical studies and in international practice guidelines [14].

In light of recent advances in the therapeutic armamentarium for AML, we have analysed the changes in first-line treatment choice in real life in Italy and have collected the opinions of experts on the potential modifications of SIE/SIES/GITMO criteria which may improve the choice of the novel strategies.

## 2. Materials and Methods

A retrospective survey has been conducted in 14 hematologic units of the Rete Ematologica Lombarda. Consecutive patients aged >65 with AML diagnosed during the period January 2020–December 2022 have been considered. Diagnoses were made according to ELN 2017 international criteria, and patients receiving antileukemic treatment were managed by the same Institutions where they had been diagnosed, either as inpatients or as outpatients, according to treatment intensity and institutional policy. The SIE/SIES/GITMO criteria were included in the initial patient work-up, and corresponding clinical and laboratory evaluation was performed accordingly, including the left ventricular ejection function (LVEF) by Echo and pulmonary function tests. Performance status was assessed by the Eastern Cooperative Oncology Group (ECOG) scale and categorized as 0–1 or 2–3 (0, fully active; 1, ambulatory; 2, in bed <50% of the time; 3, in bed >50% of the time; 4, completely bedridden). Patients addressed to BSC could also receive treatment also in non-hematological centres when logistically more appropriate. 

Clinicians have been asked to specify the type of treatment actually selected as first-line therapy for patients. Treatment modalities have been categorized as follows: (1) intensive chemotherapy (ICT), defined as any combination of cytostatic agents given with the purpose of achieving complete remission through the induction of bone marrow aplasia; (2) combination of a hypomethylating agent and Venetoclax (VEN/HMA); (3) any hypomethylating agent given as a single agent; (4) best supportive care (BSC), which included the use of hydroxyurea to control peripheral blastosis, blood cell transfusions and any ancillary treatment given to ameliorate patient’s symptoms; (5) other regimens or drugs. The results were compared with those of a previous survey conducted within the same centers in patients diagnosed in the period 2008–2016 and published by Borlenghi et al. [13].

Clinicians who were in charge of leukemic patients in the same centres convened in two meetings in 2021 and 2022 and were asked, in light of the new clinical scenario, to report their personal experience on the use of new drugs and on the modifications of the SIE/SIES/GITMO criteria which may potentially improve appropriate treatment selection for unfit patients. A final round of opinions was collected in 2023 by asking to fill out a questionnaire listing the modifications of the criteria, if any, used at their centre to select treatment with the combination VEN-HMA rather than with HMAs as a single agent. 

## 3. Results

In 529 older patients with AML at diagnosis, intensive chemotherapy was used as first-line treatment in 96 patients (18.1%). Among the 433 patients excluded from intensive chemotherapy, 247 (57%) received first-line treatment, non-intensive treatment, and 186 (43%) best supportive care. Among non-intensive therapies used, VEN/HMA was given to 167 patients (68%), while HMAs as a single agent were given to 69 (28%). Eleven patients (4%) were treated with alternative drugs, mainly within clinical studies. Given their small number and heterogeneity, they were excluded from further analyses (Figure 1). Considering the entire evaluable population, intensive chemotherapy was given to 18%, VEN and azacytidine (AZA) to 32%, HMAs as a single agent to 13%, and best supportive care to 35% of patients, respectively.

In the previous REL survey conducted from 2008 to 2016 [13], which evaluated a total of 686 patients, 274 (40%) had been treated with ICT, 139 with HMAs (19.5%), and 278 (40.5%) with BSC. Of note, during this first period, Venetoclax was not available, and HMAs were only partially available at Centers, particularly during the early times of the survey. 

Results of the present updated analysis show that the use of ICT as first-line treatment significantly decreased from 40% to 18.1% (Fisher’s exact test: *p* < 0.0001), whereas the use of HMAs with or without Venetoclax markedly increased from 19.1 to 57% (*p* < 0.0001). Since the proportion of patients receiving single-agent HMAs decreased from 19.5% to 13% (*p* < 0.005), the greater use of HMAs in recent years has been completely accounted for by the introduction of the combination VEN/HMA. The use of BSC decreased from 40.5% to 35.2% (*p* = 0.11) during the more recent period of analysis.

The modifications or integrations to SIE/SIES/GITMO criteria suggested by hematologists of 12 of 14 REL centers are summarized in Table 1.

Age limits for administering VEN-AZA are set by 11 of 12 responding centres. There was moderate disagreement since 80 years is considered the age limit by four centres and 85 years by four centres. Three centres set the limit between 80 and 85 years and consider that additional parameters should be taken into account for the choice. 

The SIE/SIES/GITMO criteria defined the presence of refractory congestive heart failure as the only limit to the use of non-intensive therapy in unfit patients, while a documented heart disease with an ejection fraction (EF) < 50% was the limit for the use of intensive therapy. All REL centres consider that a limit of the EF should be added when choosing to treat patients with VEN-HMA rather than with HMAs only. An EF% cutoff of 40% is used by eight centres, whereas three centres consider 50% and one 45% as the cutoff.

Pulmonary comorbidity excluding patients both from intensive and non-intensive chemotherapy according to SIE/SIES/GITMO, included a DLCO ≤ 65% or a FEV1 < 65% or the presence of dyspnea at rest or oxygen requirement. A total of 10 out of 12 clinicians consider that prolonged neutropenia and infection risk significantly increase the risk of pulmonary toxicity or adverse events using VEN-HMA. A thorough evaluation of the patient is therefore performed before starting treatment, often including a thorax CT scan. The presence of clinically significant COPD or an oxygen saturation at rest below 92%, particularly when associated with frequent COPD exacerbations, is used by five centres as a criterion to exclude patients from receiving the combination VEN-HMA. Further exclusion criteria are the presence of documented bronchiectasis in three centres, recurrent infections (two/year) in four centres, and colonization by multidrug-resistant microorganisms in one.

Patients on dialysis and older than 60 were to be excluded from any treatment except best supportive care according to SIE/SIES/GITMO criteria. That criteria are maintained unchanged by three centres. The age limit for patients on dialysis is raised to 70 years and to 75 years by one centre each, while in five centres the requirement of dialysis by the patient is not considered “per se” a contraindication to the use of VEN-HMA. In two centres, an estimated GFR below 30 mL/min is considered a contraindication to the use of VEN/AZA. 

According to SIE/SIES/GITMO criteria, liver comorbidity defined as cirrhosis Child B or C would exclude patients from treatment. In eight centres, the presence of Child B cirrhosis allows the delivery of Venetoclax in combination with HMAs, and in three of them have some limitations related to aminotransferase levels or to other biochemical parameters of liver function.

No modifications are applied by REL centres in the selection of patients receiving VEN-HMA in the criteria related to the domains of cognitive impairment, active infection, ECOG PS, and uncontrolled neoplasia. 

On the other hand, ten of twelve centres add to existing criteria the need for the presence of an adequate caregiver and/or the proximity of the patients to the treating centre, to allow adherence to the frequent accesses to the outpatient clinic required for diagnostic or therapeutic reasons during treatment with VEN-HMA. Further geriatric parameters, including ADL/IADL and physical performance, or being a Jehovah’s Witness are considered by two and one REL centre each. One centre underscores the opportunity of constantly re-evaluating geriatric parameters during treatment. An important limitation recalled by one centre was to preferably avoid Venetoclax when a drug interacting with Venetoclax pharmacokinetics is absolutely needed for treating a specific comorbidity in a patient. 

## 4. Discussion

The survey conducted among 14 hematologic centres that provide hematological care for about 10 million persons living in northern Italy demonstrates that the therapeutic scenario in elderly AML has markedly changed in recent years when compared to the last decade. The use of VEN-HMA has become the most frequently used first-line treatment modality. The combination has been registered for use in patients with AML deemed unfit to receive intensive chemotherapy [15]. However, the reduction in the proportion of patients receiving intensive CT, which is more than halved in the most recent survey, suggests that VEN-AZA may be currently used also in patients fit for intensive CT, owing to its marked efficacy, particularly in molecularly defined subgroups of patients, i.e., NPM-1- or IDH-mutated, as well as to the low efficacy demonstrated by chemotherapy in patients with biological high-risk prognostic features, including adverse cytogenetics.

The efficacy of VEN-HMA has been largely reported. Compared to HMAs, the combination obtains higher rates of complete remission, and a significant proportion of patients also achieve MRD eradication. Survival is significantly prolonged. The possibility of long-term disease eradication and of stopping treatment in responsive patients is currently being investigated, as well as the use of the combination in candidates to allogeneic stem cell transplantation, further indicating that its use may not be limited to unfit patients [16]. 

The toxicity profile of VEN-HMA also markedly differs from that of single-agent HMAs. The mechanism of action of VEN-HMA is only partially known. It is likely not limited to targeted bcl-2 antagonism nor to the induction of differentiation of myeloid precursors by HMAs. In vitro studies have demonstrated that VEN-HMA impairs oxidative phosphorylation through disruption of the tricarboxylic acid cycle, particularly in normal and leukemic blast cells. Hematologic toxicity induced by the combination is so deep and prolonged that it is more comparable to the effects of cytostatic agents given as combination chemotherapy than to the hematologic toxicity of HMAs. As a consequence, a higher rate of infectious complications can be expected and has been reported both in randomized trials and in the real-life experience [15,17]. In addition, the management of patients in an outpatient setting becomes more complex, and a higher number of accesses is required for patient monitoring and management. 

The increasing use of VEN-HMA combined with its unique toxicity profile dictates a reappraisal of the criteria used for evaluating the fitness of an older AML patient to the use of the combination [18]. Particularly, its uncontrolled use in unfit patients without considering an upper age limit or specific fitness parameters raises some concerns. 

Indeed, this study shows that the criteria defining patients unfit for intensive CT, proposed by SIE/SIES/GITMO when VEN-HMA was not yet available, are already currently adapted to the selection of candidates for VEN-HMA with several modifications. Clinicians who have gradually gained experience to appropriately face the problems arising during treatment have introduced warnings and limitations before starting VEN-HMAs in an old patient with AML. From the survey conducted, the most important limiting changes refer to age and cardiac and pulmonary comorbidity. 

In the SIE/SIES/GITMO criteria, no formal age limit has been set for patients fit for non-intensive treatments, including HMAs. However, this survey shows that the vast majority of REL hematologists do not consider patients older than 80–85 year old candidates for the combination of VEN-HMA. This highlights that, although licensed for patients unfit for intensive CT, the toxicity of VEN-AZA should preclude its indiscriminate use without an age limit. Further studies are needed to show if this limit should be 85 rather than 80 years. 

Likewise, according to SIE/SIES/GITMO, a refractory congestive heart failure without a specific cardiac LVEF% limitation is considered the only contraindication to non-intensive treatments. Potential cardiac toxicity of VEN-HMA has been recently reported [19], but more data are needed to assess whether this combination may induce cardiac damage. Anyway, in every REL centre, the cardiac LVEF% is among the parameters considered for selecting VEN-AZA treatment, with different limits ranging from 40% to 50%. More experience is required to confirm this prudential choice and to set a formal LVEF% limit for the use of VEN-AZA.

The presence of recurrent lung infections, bronchiectasis, or exacerbating COPD would exclude patients from the use of the combination in most centres, suggesting a safer use of single agent HMAs, if other SIE/SIES/GITMO criteria are fulfilled. 

An important addition to the criteria adopted to start treatment with VEN-HMA in older patients is the absolute need for an adequate caregiver and the possibility of adhering to the strict schedule of visits in the outpatient clinic, where treatment is delivered in most centres. The very prolonged cytopenic period that follows the use of VEN-HMA, particularly after the first cycle, and the severity of neutropenia, which puts patients at a very high risk of severe infectious complications, require that both a strict adherence to home treatment, frequent monitoring of health status and the possibility of rapid access to hospital care, are constantly guaranteed by an adequate home and family environment.

On the other hand, an extension of the use of HMAs in unfit patients, even in combination with Venetoclax, is suggested regarding renal and liver comorbidities. Patients with Child B liver cirrhosis and patients on dialysis are not a priori excluded from treatment by many clinicians, albeit with various limitations in aminotransferase levels, age, and glomerular filtration rate. The experience gained by hematologists in the use of HMAs over many years and the overall limited liver and renal toxicity of Venetoclax, provided that the ramp-up dose schedule is strictly applied, have likely led hematologists to a more expanded use of these drugs, also considering the otherwise dismal prognosis of untreated elderly AML.

The proportion of patients deemed unfit even for non-intensive treatments decreased not significantly during the most recent period, remaining however higher than 35%. Indeed, we did not expect that the introduction of a treatment modality, VEN-HMA, potentially more toxic than HMAs as a single agent, would reduce the proportion of patients receiving just the best supportive care. The survey confirms that about one-third of unselected older patients with a diagnosis of AML actually receive supportive care only. This result should be interpreted considering the real-life setting of the study, without a formal age limit, the frailty of a substantial proportion of the oldest patients, and the logistical problems raised by treatment modalities, including HMAs, which require frequent day hospital accesses and a dedicated caregiver. Two population-based retrospective studies conducted in the same period of the previous REL survey largely confirm the present data. In an analysis of elderly AML patients included in the Surveillance, Epidemiology and End Results-Medicare database, 3209 of 7665 patients (41.9%) received supportive care, only while in a non-interventional Sweden study, including 2954 AML patients aged ≥18, approximately 34% of patients received palliative treatment up-front [20,21]. On the whole, these data confirm that treatment of AML occurring in older persons still represents an unmet need for a large proportion of cases. 

The present study is limited by the lack of objective data supporting the clinical decisions adopted by hematologists. While more objectively defined fitness criteria will derive from an in-depth analysis of available studies as well as ongoing focused retrospective and prospective trials, the present study shows that hematologists are already using adapted fitness criteria to candidates for VEN-HMA therapy, according to their experience and clinical judgement. Hopefully, the modifications and warnings herein reported may help other clinicians in the difficult tailoring of AML treatment in older AML patients, trying to obtain the best balance between efficacy and safety. 

## Figures and Tables

**Figure 1 cancers-16-00386-f001:**
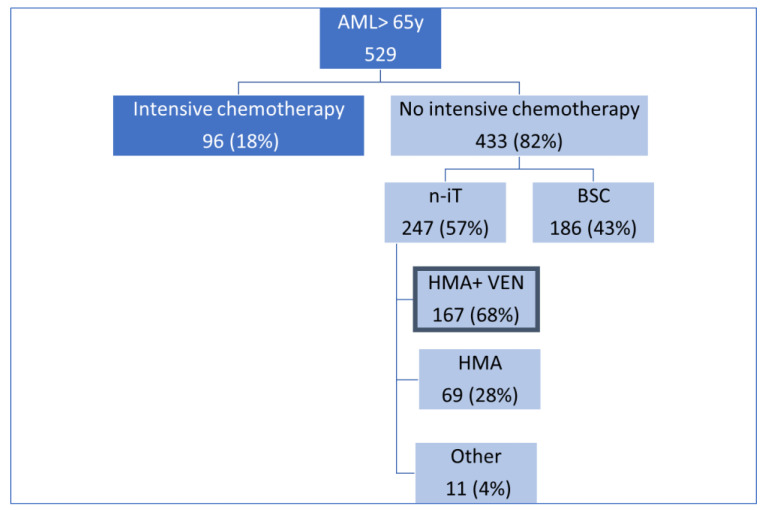
Patient disposition according to treatment received. Abbreviations: n-iT: non-intensive therapy; BSC: best supportive care, HMAs: Hypomethylating agents; VEN: Venetoclax; HMA+VEN: Hypomethylating plus venetoclax.

**Table 1 cancers-16-00386-t001:** Summary of amendments used by hematologists of the REL centers to consider treatment with VEN-HMA for older patients with AML considered unfit for intensive chemotherapy according to SIE/SIES/GITMO fitness criteria (to bold the modified Criteria).

Fitness Criteria	SIE/SIES/GITMO Criteria	REL (Rete Ematologica Lombarda) CENTERS											
		1	2	3	4	5	6	7	8	9	10	11	12
**Age Limit**	>75 y	**80 y**	**80–85 y**	**80 y**	**85 y**	**No Cut Off**	**80–85 y**	**85 y**	**85 y**	**80–85 y**	**85 y**	**80 y**	**80 y**
**PS (ECOG)**	PS > 2 not related to leukemia	NC °	NC °	NC °	NC °	NC °	NC °	NC °	NC °	NC °	NC °	NC °	NC °
**Cardiac Comorbidity**	Cardiac comorbidity (EF ≤ 50%) *	**EF 40% ****	**EF 40% *****	**EF 40%**	**EF 40%**	**EF 40%**	**EF 40%**	**EF 50%**	**EF 50%**	**EF 40%**	**EF 45%**	**EF 50%**	**EF 40%**
**Pulmonary**	Severe pulmonary comorbidity ^	**Recurrent infections**	**Bronchiectasis**	**COPD: >2 exacerbations/y**	**Bronchiectasis;** **MR colonization**	**Oxygen need**	**No infection at lung CT scan**	**COPD: frequent exacerbations**	**No limits by pulmonary function tests**	**COPD; Bronchiectasis**	**SO2 < 92%: >3 infections by y**	**COPD**	**Documented recurrent infections**
**Renal**	On dialysis and age > 60 y ^^	NC °	**eGFR > 30 mL/min**	**eGFR > 30 mL/min**	NC °	NC °	NC °	NC °	NC °	NC °	NC °	NC °	NC °
**Liver**	Severe hepatic comorbidity ^^^	NC °	**Child C or AST/ALT > 3 N**	**Child C; ** **Child B: TBE °°**	**Child C**	**Child C**	NC °	NC °	**Child C; ** **Child B: TBE °°**	**Child C**	**Child C**	NC °	**Child C**
**Cognitive** **impairment**	Current mental illness ^^^^	NC °	NC °	NC °	NC °	NC °	NC °	NC °	NC °	NC °	NC °	NC °	NC °
**Neoplasia** **uncontrolled**	Neoplasia uncontrolled	NC °	NC °	NC °	NC °	NC °	NC °	NC °	NC °	NC °	NC °	NC °	NC °
**Further** **Comorbidities**	Any other Comorbidities ^^^^^	NC °	NC °	NC °	NC °	NC °	NC °	NC °	NC °	NC °	NC °	NC °	NC °
**Social ** **domains**	not mentioned	**Absence adequate caregiver**	**Absence adequate caregiver**	**Absence adequate caregiver**	**Absence adequate caregiver**			**Absence adequate caregiver**	**Absence adequate caregiver**		**Absence adequate caregiver**	**Jehovah’s** **Witness**	**Absence adequate caregiver**
**Social ** **domains**	not mentioned		**Difficult accessibility to center §**				**Difficult accessibility to center §**		**Difficult accessibility to center §**		**Difficult accessibility to center §**	**Difficult accessibility to center §**	
**Geriatric ** **domains**	not mentioned			**ADL < 3 and/or IADL * < 4**						**Low ADL/IADL; impairment in SPPB**			

Abbreviations: y: year; PS: performance status; EF: ejection function; COPD: Chronic obstructive pulmonary disease; MR: multiresistant microorganisms; eGFR: estimated Glomerular Filtration Rate; AST: aspartate transaminase, ALT: alanine transaminase; ADL: activities of daily living; IADL: instrumental activities of daily living; SPPB: short physical performance battery. Notes: NC °: no change from SIE/SIES/GITMO criteria; TBE °°: to be evaluated on a case by case basis; * Congestive heart failure or documented cardiomyopathy with an EF ≤ 50%; ** or three episodes/year of cardiac failure requiring intravenous treatment; *** or presence of significant valvulopathy at risk of cardiac failure; ^ Documented pulmonary disease with DLCO ≤ 65% or FEV1 ≤ 65%, or dyspnea at rest or requiring oxygen, or any pleural neoplasm or uncontrolled lung neoplasm; ^^ On dialysis and age older than 60 years or uncontrolled renal carcinoma; ^^^ Liver cirrhosis Child B or C, or documented liver disease with marked elevation of transaminases (>3 times normal values) and an age older than 60 years, or any biliary tree carcinoma or uncontrolled liver carcinoma or acute viral hepatitis; ^^^^ Current mental illness: requiring psychiatric hospitalization, institutionalization or intensive outpatient management, or current cognitive status that produces dependence (as confirmed by the specialist) not controlled by the caregiver; ^^^^^ Any other comorbidity that the physician judges to be incompatible with conventional intensive chemotherapy; § because of distance from centre or too frequent visits as outpatients.

## Data Availability

Data available from the corresponding author on request due to restrictions e.g., privacy or ethical.

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
