# Peer review of "Adapting the Fitness Criteria for Non-Intensive Treatments in Older Patients with Acute Myeloid Leukemia to the Use of Venetoclax-Hypomethylating Agents Combination—Practical Considerations from the Real-Life Experience of the Hematologists of the Rete Ematologica Lombarda"

_cancers, 2024, doi:10.3390/cancers16020386_

Round 1

Reviewer 1 Report

Comments and Suggestions for Authors

Rossi et al present for review a survey of the REL group of hematologists on the treatment of patients with AML who are newly diagnosed.  The survey captures the rapid evolution of standards of care in AML which highlights the large use of HMA/Venetoclax.  The survey captures a number of metabolic and end organ functions with centers use to guide the use of HMA/Ven.  The details are largely in line with the Ferrara criteria used for inclusion in the VIALE A randomized Phase III trial of AZA/Ven.  The manuscript does capture important details which will serve as baseline for future studies and analysis. There are only a few minor topics which should be considered or mentioned before acceptance.

1)      Assessment of LVEF by Echo prior to 7&3 is a known standard of care but assessment in unfit for intensive chemotherapy is not routine in this reviewers area.  Likewise PFTs are not done routinely for either therapy.  The authors are encouraged to mention if these test are routinely done before therapy in REL group or is therapy modified only if the ECHO or PFTs are known prior to leukemia diagnosis.

2)      The authors mention no change in ECOG PS criteria over the two study periods but it may be helpful to clarify the performance status criteria as these are universally detailed in study reports

3)      The large group of patients treated with BSC in this study population with aza/ven now available highlights areas of unmet need and future studies.  In the discussion it may be helpful for the authors to summarize their opinions on why 35% of over age 65 patients are still receiving BSC. 

Comments on the Quality of English Language

Good.

Author Response

Rossi et al present for review a survey of the REL group of hematologists on the treatment of patients withAML who are newly diagnosed. The survey captures the rapid evolution of standards of care in AML which highlights the large use of HMA/Venetoclax. The survey captures a number of metabolic and end organ functions with centers use to guide the use of HMA/Ven. The details are largely in line with the Ferrara criteria used for inclusion in the VIALE A randomized Phase III trial of AZA/Ven. The manuscript does capture important details which will serve as baseline for future studies and analysis. There are only a few minor topics which should be considered or mentioned before acceptance.

  • Assessment of LVEF by Echo prior to 7&3 is a known standard of care but assessment in unfit for intensive chemotherapy is not routine in this reviewers area. Likewise PFTs are not done routinely for either therapy. The authors are encouraged to mention if these test are routinely done before therapy in REL group or is therapy modified only if the ECHO or PFTs are known prior to leukemia diagnosis.

The LVEF by Echo and PFT parameters are included in the SIE/SIES/GITMO criteria which are routinely applied by REL Centers to help in treatment choice for older patients with AML. This has been clarified by adding a sentence in the M&M section (line 113), highlighted in yellow.

  • The authors mention no change in ECOG PS criteria over the two study periods but it may be helpful to clarify the performance status criteria as these are universally detailed in study reports

We have added the criteria used fo the assessment of PS by the ECOG scale in full in the M&M section (line 116), highlighted in yellow.

  • The large group of patients treated with BSC in this study population with aza/ven now available highlights areas of unmet need and future studies. In the discussion it may be helpful for the authors to summarize their opinions on why 35% of over age 65 patients are still receiving BSC.

We thank the reviewer for his thoughtful comment which highlights the still important difficulties in the selection, in the real life, of older AML patients candidates even to non-intensive treatments. We believe that the real life context of the study the lack of any upper age limit as well as the logistical problems related to the delivery of HMA are the most important causes of the apparently high proportion of BSC patients in our retrospective review. We have added a paragraph to the discussion to highlight this concepts (line 272, highlighted in yellow) and have reported the results of two studies whose results are similar to those of our series, adding them to the references.

Reviewer 2 Report

Comments and Suggestions for Authors

I have read the paper investigating patterns of care for elderly patients with AML in Italy with great interest. I present here the remarks aimed at further improving this very interesting manuscript. 

1) The paper refers to previous REL survey, it would be more readable if authors provide the period that previous survey relates to in the same sentence. 

2) Table 1 in the manuscript looks like copy-paste table from excel still retaining formatting and comments from the original program. This should be corrected. 

3) There is incomplete information on full criteria, as table 1 states no change from previous criteria. This is crucial to be presented (full criteria per center) to better understand differences in delivery of care.  

4) Table 2 providing insight into proportion of different therapies per same centers would be of interest (to see how strictness in these criteria affect proportions of specific therapies delivered). 

Author Response

I have read the paper investigating patterns of care for elderly patients with AML in Italy with great interest. I present here the remarks aimed at further improving this very interesting manuscript. 

  • The paper refers to previous REL survey, it would be more readable if authors provide the period that previous survey relates to in the same sentence.

The study period has been added to the text (line 146) highlighted in yellow.

  • Table 1 in the manuscript looks like copy-paste table from excel still retaining formatting and comments from the original program. This should be corrected. 

Table 1 has been corrected according to the suggestion

3) There is incomplete information on full criteria, as table 1 states no change from previous criteria. This is crucial to be presented (full criteria per center) to better understand differences in delivery of care.

The SIE/SIES/GITMO criteria used without changes by the Centers have been added to Table 1 which has been further expanded by reporting the SIE/SIES/GITMO criteria as defined in the original paper. We hope that this amendment have improved the readability of the Table.

4) Table 2 providing insight into proportion of different therapies per same centers would be of interest (to see how strictness in these criteria affect proportions of specific therapies delivered). 

We thank the Reviewer for raising this interesting point. However clinical data on a patient by patient basis were not available for this study whose main aim was to analyse the hematologist’s attitude in the selection of recently available treatments for older AML patients. REL centers are conducting a more detailed analysis on a single-patient basis to investigate on this and other clinical questions (preliminary results have been presented as oral communication at last meeting ASH 2023: Borlenghi et al: “SIE/SIES/GITMO Criteria in Elderly Patients with Acute Myeloid Leukemia (AML): Useful also in the New Drug Era? A Multicentric Real-Life Study by the hematological network Rete Ematologica Lombarda (REL).” Blood 2023, 142 (supplem 1): 956)

Reviewer 3 Report

Comments and Suggestions for Authors

In this manuscript, the Authors have described reasons underlying therapeutic choices in acute myeloid leukemia (AML) older patients (Aged >65 years old) followed by Centers in the Italian "Rete Ematologica Lombarda" over a 2-year period and compared to an historical cohort (2008-2016).

However, this poses a bias as patients in the historical cohort have been stratifed with different criteria (not the ELN 2017 used in studied patients).

The paper should be reorganized as a brief report rather than a complete research article, and title should be rephrased as Authors described more a real-life experience in the Northern Italia rather than exploring practical considerations or fitness assessment.

The Authors should also indicated when and how venetoclax was given (first line? How many patients as second line treatment?).

As per technical data sheet, there are no age limits; therefore, the Authors should discuss this choice, as well as this heterogeneity in ejection fraction cut-off values.

Finally, as fitness assessment has been claimed in the title, no scales or scoring systems were displayed or described in the analysis.

Author Response

In this manuscript, the Authors have described reasons underlying therapeutic choices in acute myeloid leukemia (AML) older patients (Aged >65 years old) followed by Centers in the Italian “Rete Ematologica Lombarda”; over a 2-year period and compared to an historical cohort (2008-2016).

  • However, this poses a bias as patients in the historical cohort have been stratified with different criteria (not the ELN 2017 used in studied patients). The paper should be reorganized as a brief report rather than a complete research article, and title should be rephrased as Authors described more a real-life experience in the Northern Italia rather than exploring practical considerations or fitness assessment.

We thank the Reviewer for this comment and agree that ELN treatment recommendations were not available before 2017. They stratify patients according to their biological risk. However our survey was not intended to capture treatment choices according to the biological risk of AML. We only report the treatments actually used in older AML patients within REL centers, showing that, as expected, the recent availability of VEN-AZA has markedly changed the therapeutic scenario. A more in depth analysis correlating treatment choices with other clinical parameters is underway and will be available hopefully in 2024. The aim of the study was to report on the attitude of REL hematologists towards the choice of VEN-AZA according to the fitness of older AML patients. As above mentioned, the available clinical data were limited to treatment choices, which imply that clinicians have preliminary evaluated the fitness of their patients either to intensive CT or non-intensive treatments, including VEN-HMA, or just to best supportive care

  • The Authors should also indicated when and how venetoclax was given (first line? How many patients as second line treatment?).

As reported in the Results section (line 136), the paper refers only to patients at diagnosis and to the first-line treatments. To avoid any misunderstanting we have integrated the text (line 139, highlighted in yellow).

  • As per technical data sheet, there are no age limits; therefore, the Authors should discuss this choice, as well as this heterogeneity in ejection fraction cut-off values

The paper focuses on the potential modifications of SIE/SIES/GITMO criteria in clinical practice after the introduction of treatment modalities (VEN-HMA) not available during the first survey period. Since no age limits were established by SIE/SIES/GITMO criteria for the category of patients deemed unfit to intensive chemotherapy but not unfit to non-intensive treatments we did not set any age limit. To more extensively discuss the age and the cardiac LVEF% limits used by REL hematologists two specific paragraphs and one reference have been added to the discussion (Line 240 highlighted in yellow).

  • Finally, as fitness assessment has been claimed in the title, no scales or scoring systems were displayed or described in the analysis

The treatment-oriented fitness criteria used in the survey were those proposed by SIE/SIES/GITMO in 2013. They have been reported in reference 11 and have been further described in Table 1, according to the suggestion also of Reviewer 2. If it is considered mandatory we may add to the paper a Table displaying these criteria.

Round 2

Reviewer 2 Report

Comments and Suggestions for Authors

Thank You, the review is satisfactory 

Author Response

Thank you

Reviewer 3 Report

Comments and Suggestions for Authors

The Authors have addressed all comments; however, it might be worth to reconsider the title, as already suggested, because in the present form is misleading.

Author Response

Thank you for your suggestion. We could change the title like this: “Adapting the fitness criteria for non-intensive treatments in older patients with acute myeloid leukemia to the use of venetoclax-hypomethylating agents combination. Practical considerations from the real-life experience of the hematologists of the Rete Ematologica Lombarda”